# Cognitive Performance and Health-Related Quality of Life in Patients with Neuromyelitis Optica Spectrum Disorder

**DOI:** 10.3390/jpm12050743

**Published:** 2022-05-02

**Authors:** Elisabet Lopez-Soley, Jose E. Meca-Lallana, Sara Llufriu, Yolanda Blanco, Rocío Gómez-Ballesteros, Jorge Maurino, Francisco Pérez-Miralles, Lucía Forero, Carmen Calles, María L. Martinez-Gines, Inés Gonzalez-Suarez, Sabas Boyero, Lucía Romero-Pinel, Ángel P. Sempere, Virginia Meca-Lallana, Luis Querol, Lucienne Costa-Frossard, Maria Sepulveda, Elisabeth Solana

**Affiliations:** 1Center of Neuroimmunology, Laboratory of Advanced Imaging in Neuroimmunological Diseases, Hospital Clinic Barcelona, Institut d’Investigacions Biomediques August Pi i Sunyer (IDIBAPS) and Universitat de Barcelona, 08036 Barcelona, Spain; elopez2@clinic.cat (E.L.-S.); sllufriu@clinic.cat (S.L.); yblanco@clinic.cat (Y.B.); msepulve@clinic.cat (M.S.); 2Department of Neurology, Clinical Neuroimmunology Unit and Multiple Sclerosis CSUR, Hospital Universitario “Virgen de la Arrixaca”, IMIB-Arrixaca, 30120 Murcia, Spain; pmecal@gmail.com; 3Medical Department, Roche Farma, 28042 Madrid, Spain; rocio.gomez@roche.com (R.G.-B.); jorge.maurino@roche.com (J.M.); 4Department of Neurology, Unit of Neuroimmunology, Hospital Universitari i Politècnic La Fe, 46026 Valencia, Spain; miralles_neuro@hotmail.com; 5Department of Neurology, Hospital Universitario Puerta del Mar, 11009 Cadiz, Spain; lucia.forero.diaz@hotmail.com; 6Department of Neurology, Hospital Universitari Son Espases, 07120 Palma de Mallorca, Spain; mcalles22@yahoo.es; 7Department of Neurology, Hospital Universitario Gregorio Marañón, 28007 Madrid, Spain; marisamgines@hotmail.com; 8Department of Neurology, Hospital Universitario Álvaro Cunqueiro, 36213 Vigo, Spain; igonsua@gmail.com; 9Department of Neurology, Hospital Universitario Cruces, 48903 Bilbao, Spain; sabasboyero@gmail.com; 10Department of Neurology, Hospital Universitari de Bellvitge, 08907 Barcelona, Spain; lromeropinel@gmail.com; 11Department of Neurology, Hospital General Universitario de Alicante, 03010 Alicante, Spain; aperezs@mac.com; 12Department of Neurology, Hospital Universitario La Princesa, 28006 Madrid, Spain; virmeca@hotmail.com; 13Department of Neurology, Hospital de la Santa Creu i Sant Pau, 08025 Barcelona, Spain; lquerol@santpau.cat; 14Department of Neurology, Hospital Universitario Ramón y Cajal, 28034 Madrid, Spain; lufrossard@yahoo.es

**Keywords:** neuromyelitis optica spectrum disorder, cognition, health-related quality of life, mood

## Abstract

Background: The frequency of cognitive impairment (CI) reported in neuromyelitis optica spectrum disorder (NMOSD) is highly variable, and its relationship with demographic and clinical characteristics is poorly understood. We aimed to describe the cognitive profile of NMOSD patients, and to analyse the cognitive differences according to their serostatus; furthermore, we aimed to assess the relationship between cognition, demographic and clinical characteristics, and other aspects linked to health-related quality of life (HRQoL). Methods: This cross-sectional study included 41 patients (median age, 44 years; 85% women) from 13 Spanish centres. Demographic and clinical characteristics were collected along with a cognitive z-score (Rao’s Battery) and HRQoL patient-centred measures, and their relationship was explored using linear regression. We used the Akaike information criterion to model which characteristics were associated with cognition. Results: Fourteen patients (34%) had CI, and the most affected cognitive domain was visual memory. Cognition was similar in AQP4-IgG-positive and -negative patients. Gender, mood, fatigue, satisfaction with life, and perception of stigma were associated with cognitive performance (adjusted R^2^ = 0.396, *p* < 0.001). Conclusions: The results highlight the presence of CI and its impact on HRQoL in NMOSD patients. Cognitive and psychological assessments may be crucial to achieve a holistic approach in patient care.

## 1. Introduction

Neuromyelitis optica spectrum disorder (NMOSD) is an inflammatory autoimmune disorder of the central nervous system (CNS) predominantly targeting the spinal cord and optic nerve [1,2]. The discovery of an immunoglobulin G directed against the astrocyte water channel protein aquaporin-4 (AQP4-IgG) not only allowed a reliable distinction of the disease from multiple sclerosis (MS), the most common differential diagnosis [3], but also led to expansion of the clinical syndromes associated with the disorder and the definition of a new set of diagnostic criteria with prognostic implications (2015 criteria) [4].

Most NMOSD patients follow a course of early disability accrual due to frequent and potentially severe relapses. In recent years, increasing attention has been paid to the prevalence and pattern of cognitive impairment (CI) in NMOSD patients, as it is an underestimated but disabling symptom with imprecise description [5]. The frequency of CI varies substantially across studies, ranging from 3% to 75% [6,7], with methodological heterogeneity in terms of samples enrolled, diagnostic criteria applied, CI definition or the neuropsychological assessment tools employed [1,7,8]. Previous studies not only have high variations across the frequency of CI, but also depict ambiguous results about the most affected cognitive domains in NMOSD patients. Moreover, it is not entirely clear whether the presence or absence of AQP4-IgG could influence cognitive performance.

Other aspects related to the disease, such as mood, fatigue, and self-perception of symptoms and pain have an impact on the patient’s quality of life, interfering with physical and emotional aspects of wellbeing [9,10]. However, the relationship of these factors with NMOSD patients’ cognitive performance has been poorly investigated. A further analysis of the full spectrum of cognitive performance and the impact of psychological comorbidities is needed for a better understanding of the disease’s symptoms, and to provide potential target interventions. Therefore, the main objective of this study was to describe the cognitive profile of a well-characterised group of patients with NMOSD, and to analyse cognitive differences according to their serostatus. The secondary objective was to assess the relationship between cognition, demographic and clinical characteristics, and the contribution of emotional status and other aspects related to the health-related quality of life (HRQoL).

## 2. Materials and Methods

### 2.1. Participants

For this non-interventional cross-sectional study, we collected data from patients consecutively recruited at thirteen hospital-based neuroimmunology clinics in Spain (PERSPECTIVES-NMO study) [11] between November 2019 and July 2020. The inclusion criteria were (a) patients aged between 18 and 65 years; (b) diagnosed with NMOSD according to the Wingerchuk 2015 criteria [4]; (c) relapse-free or not having received corticosteroids in the last 30 days; (d) stable treatment in the last three months and; (e) available cognitive and mood disorder assessments. Patients with difficulties in understanding and/or responding to the study questionnaires and with other concomitant chronic disorders that could significantly affect cognition or mood were excluded from the study.

Thus, a total of 41 NMOSD patients fulfilled the inclusion criteria and were analysed. Epidemiological and clinical data (age, gender, educational level, disease duration, presence of AQP4-IgG antibodies, number of relapses, and current treatment) were recorded in an electronic case report form specially designed for this study. Neurological disability was assessed by the Expanded Disability Status Scale (EDSS) score [12]. We evaluated mood disorders using the Beck Depression Inventory-Fast Screen (BDI-FS) [13], with a total score ranging from 0 to 21. Higher scores indicate more severe depression symptoms with cut-off scores ≥4, ≥9, and >12 indicating mild, moderate, and severe depression, respectively. Daily fatigue was assessed by the Fatigue Impact Scale for Daily Use (D-FIS) [14], an 8-item self-report instrument in which higher scores indicate a greater impact of fatigue. The neuropsychological battery and the patient-centred measures employed are described in subsequent sections.

The study was approved by the investigational review board of Galicia (CEIm-G, Santiago de Compostela, Spain) and signed informed consent was obtained from all patients prior to their enrolment.

### 2.2. Cognitive Functions

We assessed cognitive performance using the Brief Repeatable Battery of Neuropsychological tests (BRB-N) [15]. This battery includes several tests assessing cognitive domains: (1) verbal memory: Selective Reminding Test (SRT, with two subtests: consistent long-term retrieval as an indicator of consolidation, and delayed recall); (2) visual memory: 10/36 Spatial Recall Test (SPART, with two subtests: immediate retrieval and delayed recall); (3) attention and information processing speed (IPS): Symbol Digit Modalities Test (SDMT) and Paced Auditory Serial Addition Test (PASAT) with three second per digit version; and (4) semantic fluency and cognitive flexibility: Word List Generation (WLG).

Raw values were transformed into z-scores by adjusting for age and educational level according to the available Spanish normative data [16], and then grouped in terms of global cognition (zBRB-N) and for each cognitive domain. Failure in any test was considered when z-score was below −1.5 standard deviations (SDs) of the norm. CI in a given cognitive domain was defined as a failure in at least one test assessing that domain, and global CI was defined as an impairment in at least two cognitive tests evaluating the same or different cognitive domains. Patients without global CI were categorised as cognitively preserved (CP).

### 2.3. Patient-Centred Measures

Measures of HRQoL were evaluated using the physical and psychological components of the Multiple Sclerosis Impact Scale (MSIS-29v2) [17], a self-reported questionnaire ranging from 0 to 100 with higher scores indicating worse health, and by the Satisfaction with Life Scale (SWLS) [18], a five-item measure of self-rated assessment of subjective wellbeing scored from 5 (worst) to 35 (best). Symptom severity from the patient perspective was assessed by the SymptoMScreen questionnaire (SyMS), consisting of 12 items with higher scores indicating more severe symptom endorsement [19]. The Stigma Scale for Chronic Illness 8-item version (SSCI-8) [20] was used to evaluate internalised and experienced stigma across neurological conditions. It is composed of eight items and scores range from 0 to 40 with higher scores indicating higher levels of perceived stigma. Finally, the MOS Pain Effects Scale (PES) [21] is a 6-item self-report questionnaire assessing how pain and unpleasant sensations affect mood, capacity to walk or move, sleep, work, recreation, and pleasure of life. Total score ranges from 6 to 30, with higher results suggesting greater impact of pain.

### 2.4. Statistical Analysis

We described demographic, clinical, cognitive and patient-centred measures data by the median and interquartile range (IQR) for continuous variables and by absolute numbers and relative frequencies for categorical data. The normality assumption was checked by histograms and Shapiro–Wilk test. We explored differences in demographic, clinical and cognitive characteristics between AQP4-IgG-positive and -negative NMOSD patients using the Chi-squared and Wilcoxon–Mann–Whitney *U*-test or Student’s *t*-test, when necessary, and demographic and clinical characteristics between CP and CI patients. Differences between patient-centred measures in previous groups were explored with analysis of variance.

We used linear regression to analyse the association between the z-score of global cognition (zBRB-N) and demographic (age and gender), clinical (disease duration, presence of AQP4-IgG antibodies, EDSS score, number of relapses before study inclusion, current treatment, BDI-FS and D-FIS scores), and patient-centred measures (MSIS-29v2, SWLS, SyMS, SSCI-8 and PES scores). We then fitted a multiple regression model including all the variables mentioned. We used the Akaike Information Criterion (AIC) to select the variables that best fit a model based on the whole cohort. For easier interpretation, all variables were standardised using the mean and SD.

In all analyses, we included age and gender as covariates to control for their potential influence on results. We used the false discovery rate (FDR) to correct for multiple comparisons, and we set the significance level to *p* < 0.05. All the statistical analyses were performed with R statistical software (version 3.6.0, www.R-project.org; accessed on 1 September 2021).

## 3. Results

### 3.1. Demographic, Clinical and Patient-Centred Measures of the Cohort

The demographic, clinical and patient-centred measures data of the 41 patients are summarised in Table 1. Patients were more frequently female (85%) and middle-aged (median of 44 years, IQR: 39–49), with a median disease duration of 8.1 years (IQR: 3.9–15.5) and a median EDSS score of 2.0 (range 0–7.5). Depressive symptoms were present in 18 (44%) patients: 12 (29%) had mild depression and 6 (15%) moderate depression. Four had concomitant disorders, one was also diagnosed with Sjogren’s syndrome and three more with Lupus.

Twenty-seven patients (66%) were AQP4-IgG positive. The demographic, clinical and patient-centred measures data were not significantly different between AQP4-IgG-positive and -negative patients (see Appendix A).

### 3.2. Cognitive Characteristics in NMOSD Patients

Fourteen patients (34%) were classified as having global CI. Demographic and clinical characteristics were similar (*p* > 0.05) between patients regardless of their cognitive status. However, patients with global CI had lower satisfaction with life, more severe symptom endorsement, higher levels of perceived stigma, and greater impact of pain interfering with their lives than CP patients (Appendix A).

Figure 1A summarises the cognitive z-score distribution of each test from the BRB-N. Based on the definition of CI described above, the following frequencies of impairment in each cognitive domain were recorded: 10 patients (24%) in verbal memory, 14 patients (34%) in visual memory, 13 patients (32%) in attention-IPS and 3 patients (7%) in semantic fluency (Figure 1B).

When we analysed whether cognition was similar in AQP4-IgG-positive and -negative patients, we found no statistically significant differences in either the individual test z-scores of the BRB-N or the cognitive domains (see Table 2).

### 3.3. Association between Cognition, Demographic, Clinical and Patient-Centred Measures

The global BRB-N z-score was associated with fatigue (D-FIS score: β = −0.322, 95% confidence interval, CI: −0.53, 0.12: corrected *p* = 0.013), physical impact of the disease on quality of life (MSIS-29v2: β = −0.31, 95% CI: −0.53, −0.09: corrected *p* = 0.028), satisfaction with life (SWLS: β = 0.302, 95% CI: 0.09, 0.51: corrected *p* = 0.024), self-perception of symptoms (SyMS: β = −0.327, 95% CI: −0.55, −0.11: corrected *p* = 0.019) and perception of stigma (SSCI-8: β = −0.322, 95% CI: −0.53, −0.12: corrected *p* = 0.012). Depression score was not related to cognitive performance (BDI-FS: β = −0.188, 95% CI: −0.41, 0.04: corrected *p* = 0.306).

Based on the AIC, the final multiple linear regression model included gender as well as depression (BDI-FS) and fatigue (D-FIS) scores, satisfaction with life and perception of the stigma (SWLS and SSCI-8). In our sample, 40% of the variability of the z-score of BRB-N was explained by this model (adjusted R^2^ = 0.396, *p* < 0.001). A change of 1 point in the BDI-FS questionnaire, sensitive to depression, was associated with change of 0.6 points in global cognitive scores. Fatigue (D-FIS score), satisfaction of life questionnaire (SWLS) and perception of stigma for neurological diseases (SSCI-8) were also related to cognition (Table 3).

## 4. Discussion

This study of a well-characterised cohort of patients with NMOSD diagnosed by the 2015 criteria shows that up to 34% of the patients suffer CI. Visual memory was the main cognitive domain affected, followed by attention-IPS and verbal memory. AQP4-IgG-positive and -negative NMOSD patients did not differ in their cognitive performance, despite having similar demographic and clinical characteristics. The study also identifies depression, fatigue, satisfaction with life and perception of stigma as the main factors related to global cognitive performance.

Although some attempts have been made to describe the cognitive profile in NMOSD patients, both the reported CI prevalence and the affected cognitive domains varied widely. Our results are in agreement with other studies reporting that around 34% of the patients can be classified as having CI [22,23]. However, the proportion of patients suffering impairment in our study differs from others with smaller cohorts [24,25,26], which applied different criteria for CI [6] or used other neuropsychological tools for cognitive assessment [27]. The most affected domain in our cohort was visual memory, followed by attention-IPS and verbal memory. These findings are in line with two recent reviews where memory, attention, and IPS are the most affected cognitive functions [5,8]. Similarly, Zhang and et al. found that both memory and IPS were more severely impaired in the visual than in the verbal spectrum [28]. Conversely, our results show a relatively preserved performance for semantic verbal fluency, which is one of the most pronounced dysfunctions in other studies [6,24].

We did not find an influence of clinical worsening, as measured by the number of relapses, disease duration and EDSS score, on cognitive performance. Moreover, no association was found between a positive AQP4-IgG status and cognitive performance, supporting the results of other studies exploring differences in cognitive test scores and APQ4-IgG status [28,29,30]. APQ4-IgG appears to inhibit neuronal plasticity, impacting the proper functioning of the glutamatergic system and water homeostasis by increasing excitotoxicity in cerebral grey matter [25]. However, this would not explain the CI observed in NMOSD patients who are AQP4-IgG negative. It is also unknown what causes the humoral immune response that produces the AQP4-IgG antibodies. Some infectious agents, even silent infections (*Mycobacterium avium* subspecies), have been involved in NMOSD aetiology [31,32]. Molecular mimicry between microbes and host peptides has been proposed as a mechanism that would exacerbate autoimmunity and generate autoantibodies. Interestingly, one recent study has shown a different pattern of humoral-driven immune responses against viral agents (HERV-W retroviruses family) between patients with NMOSD compared to patients with MS or MOG-IgG [33]. If such infectious agents could influence cognitive performance and its implication in autoimmunity deserve further studies. Additionally, the use of techniques such as non-conventional neuroimaging can shed light on the underlying mechanisms of cognitive decline in patients with NMOSD. In this regard, the presence of brain lesions at sites of high AQP4 expression, atrophy of deep grey matter structures or impairment of white and grey matter integrity have been proposed to be related to cognitive deficits in NMOSD [22,34]. It should be noted that the pathophysiological substrate of CI in patients with NMOSD is still not completely understood and should be further explored.

Mood disorders and fatigue are other major symptoms described in patients with NMOSD. We found a moderate association between fatigue and lower cognitive performance, while depression was not related to cognition. However, when we included fatigue and depression in the same model (after applying the AIC), among other variables related to HRQoL and gender, we found a strong correlation between depression and cognitive performance, suggesting a relationship between patients’ psychological wellbeing and their performance on cognitive tasks. The relationship between depression, fatigue and cognition is not straightforward [1,29], but the current results indicate that the combination of both factors exerts a more deleterious effect on cognitive function. Overall, these findings highlight the importance of considering depression and fatigue symptoms in patients with NMOSD in the clinical setting.

Importantly, we found differences in the patient-centred measures between patients with impaired cognition and those with preserved performance. Indeed, in our cohort, we observed that patients with global CI had lower life satisfaction, showed more severe symptom endorsement, and perceived more stigma and pain. Moreover, when we analysed the association between patient-centred measures and cognitive performance in the whole cohort, we found that the global cognitive score was associated with the physical impact of the disease on quality of life, satisfaction with life, self-perception of symptoms and perception of stigma. These findings highlight the impact of cognitive and psychological impairment on the wellbeing of NMOSD patients.

This study has some limitations. First, the cross-sectional design did not allow us to assess the dynamics of the cognitive profile in NMOSD patients. Similarly, causal relationships between cognition and patient-centred measures could not be identified, and we were not able to add any pathological aspects related to brain damage in the linear regression analysis. In addition, although our cohort of patients with NMOSD is not very large, it is similar in size to other studies in this field and influenced by the low prevalence of the disease [29,35]. Further studies including more patients will be needed to explore the cognitive profile and the influence of clinical and pathological aspects on cognition. Nevertheless, our study also has several strengths. We described cognitive performance and its relationship with demographic and clinical characteristics and patient-centred measures in a sample of patients treated across 13 different hospitals throughout Spain, allowing results to be generalised to clinical practice.

To conclude, about 34% of patients with NMOSD included in our study had cognitive dysfunction, with visual learning and memory and attention-IPS being the most affected cognitive domains. Cognition was mainly associated with mood, fatigue, and the patient’s positive attitude toward life and their perception of the disease. Cognitive and psychological assessments may be crucial to achieve a holistic approach in NMOSD patient care.

## Figures and Tables

**Figure 1 jpm-12-00743-f001:**
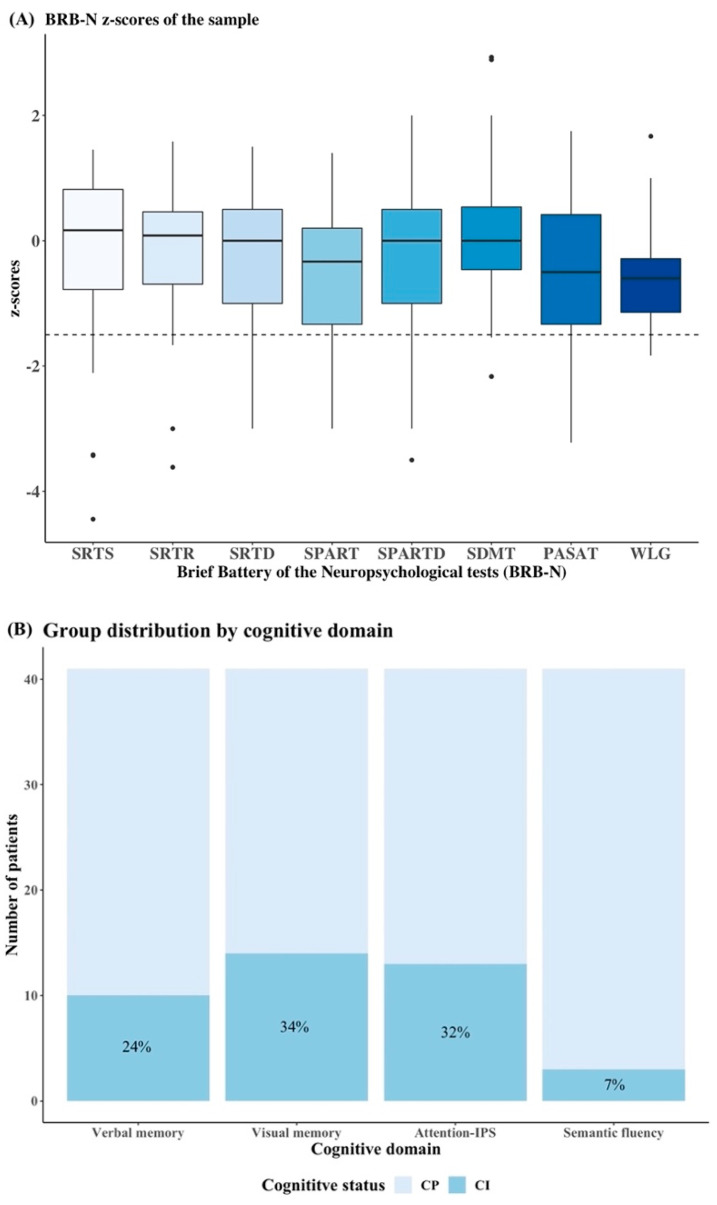
Cognitive performance in NMOSD patients. (**A**) Box plots represent the cognitive z-score distribution for each test from the BRB-N in the entire cohort; the *x*-axis depicts the name of each cognitive test and the *y*-axis the z-score for each test. The dotted black horizontal line represents −1.5 SDs of the norm. (**B**) The histograms show the proportions of patients with CP and CI in each cognitive domain. The *x*-axis shows the names of cognitive domains and the *y*-axis the number of patients for each domain. The total number of patients in each cognitive domain was 41. Both figures were fitted using R version 3.5.2 (R Foundation for Statistical Computing, Vienna, Austria). SRTS: Selective Reminding Test Long-Term Storage; SRTR: Selective Reminding Test Consistent Long-Term Retrieval; SRTD: Selective Reminding Test Total Delay; SPART: Spatial Recall Test; SPARTD: Spatial Recall Test Delay; SDMT: Symbol Digit Modalities Test; PASAT: Paced Auditory Serial Addition Task; WLG: Word List Generation.

**Table 1 jpm-12-00743-t001:** Demographic, clinical and patient-centred measures data of the study population.

	NMOSD Cohort (*n* = 41)
**Demographic and clinical data**	
Age (years)	44 (39–49)
Female, *n* (%)	35 (85)
Disease duration (years)	8.1 (3.9–15.5)
AQP4-IgG positive, *n* (%)	27 (66)
EDSS score (range)	2.0 (0–7.5)
Number of relapses	2.5 (1–4)
Current treatment, *n* (%)	37 (90)
Beck Depression Inventory-Fast Screen (BDI-FS)	3 (0–6)
Fatigue Impact Scale for Daily Use (D-FIS)	6 (2–18)
**Patient-centred measures**	
Physical MSIS-29v2	35 (23–49)
Psychological MSIS-29v2	21 (14–29)
Satisfaction with Life Scale (SWLS)	21 (18–25)
SymptoMScreen questionnaire (SyMS)	16 (8–30)
Stigma Scale for Chronic Illness (SSCI-8)	9 (8–14)
MOS Pain Effects Scale (PES)	15 (9–20)

Qualitative data are presented by absolute numbers and proportions, and quantitative data by the median and IQR, unless otherwise specified. NMOSD: neuromyelitis optica spectrum disorder; AQP4-IgG: aquaporin-4 immunoglobulin G; EDSS: Expanded Disability Status Scale; MSIS-29v2: Multiple Sclerosis Impact Scale.

**Table 2 jpm-12-00743-t002:** Cognitive performance differences between AQP4-IgG-positive and -negative patients.

Cognitive z-Score	AQP4-IgG Positive (*n* = 27)	AQP4-IgG Negative (*n* = 13)	Corrected *p*-Value
**Verbal memory**			
z-score SRTS (storage)	0.17 (−0.49–0.86)	0.17 (−0.89–0.54)	0.952 ^b^
z-score SRTR (retrieval)	0.23 (−0.60–0.50)	−0.25 (−1.08–0.46)	0.952 ^b^
z-score SRTD (delayed)	0.0 (−1.25–0.50)	−0.50 (−1.0–1.0)	0.952 ^b^
Verbal memory z-score	0.05 (−0.54–0.66)	−0.04 (−0.62–0.69)	0.977 ^b^
**Visual memory**			
z-score SPART (storage)	−0.50 (−1.42–0.55)	−0.20 (−0.67–0.17)	0.952 ^a^
z-score SPARTD (delayed)	0.0 (−1.5–0.5)	−0.50 (−1.0–0.0)	0.952 ^a^
Visual memory z-score	−0.67 (−1.40–0.55)	−0.33 (−0.67–0.08)	0.952 ^a^
**Attention and information processing speed**			
z-score SDMT	0.12 (−0.61–0.54)	0.0 (−0.29–1.29)	0.952 ^a^
z-score PASAT 3	−0.50 (−1.29–0.65)	−0.44 (−1.33–0.33)	0.952 ^a^
Attention-IPS z-score	−0.28 (−1.06–0.50)	−0.15 (−0.89–0.70)	0.952 ^a^
**Semantic fluency**			
z-score WLG	−0.60 (−0.86–−0.18)	−0.6 (−1.17–−0.29)	0.952 ^b^
Semantic fluency z-score	−0.60 (−0.86–−0.18)	−0.6 (−1.17–−0.29)	0.952 ^b^
**Global cognition (zBRB-N)**			
BRB-N z-score	−0.32 (−0.93–0.21)	−0.17 (−0.97–0.15)	0.977 ^a^

The data represent the median and IQR. AQP4-IgG: aquaporin-4 immunoglobulin G; SRTS: Selective Reminding Test Long-Term Storage; SRTR: Selective Reminding Test Consistent Long-Term Retrieval; SRTD: Selective Reminding Test Total Delay; SPART: Spatial Recall Test; SPARTD: Spatial Recall Test Delay; SDMT: Symbol Digit Modalities Test; PASAT: Paced Auditory Serial Addition Task; IPS: information processing speed; WLG: Word List Generation; BRB-N: Brief Repeatable Battery of Neuropsychological tests. The *p*-values were corrected by FDR adjustment. One patient was excluded due to unknown serostatus. ^a^ Student’s *t*-test; ^b^ Kruskal–Wallis test.

**Table 3 jpm-12-00743-t003:** Associations between the z-score of the global cognitive score (zBRB-N) and demographic, clinical and patient-centred measures.

Parameters	β (95% CI)	Corrected *p*-Value
Gender	−0.418 (−0.92–0.09)	0.102
Beck Depression Inventory-Fast Screen (BDI-FS)	0.654 (0.26–1.05)	0.002
Fatigue Impact Scale for Daily Use (D-FIS)	−0.388 (−0.72–−0.05)	0.024
Satisfaction with Life Scale (SWLS)	0.343 (0.08–0.60)	0.011
Stigma Scale for Chronic Illness (SSCI-8)	−0.361 (−0.65–−0.07)	0.016

Beta coefficients and 95% confidence intervals (CI) and *p*-values corrected by FDR adjustment.

## Data Availability

Qualified researchers may request access to individual patient-level data through the corresponding author. The datasets generated during the analysis of the study are available from the corresponding author on reasonable request.

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
