# Peer review of "Cognitive Performance and Health-Related Quality of Life in Patients with Neuromyelitis Optica Spectrum Disorder"

_jpm, 2022, doi:10.3390/jpm12050743_

Round 1

Reviewer 1 Report

  1. The sample size is too low.
  2. As authors mentioned there are some articles related to this issue. What is the novelty of this study?
  3. There are some points which have not been considered in inclusion criteria: a- psychiatric comorbidity b- any other cognitive disorders c- recent history of steroid prescription d- presence or absence of other autoimmune diseases e- recent history of relapse
  4. What is the exclusion criteria?
  5. What is the history of concomitant disorders?
  6. The seropositive patients have more severe disease. Why did not they differ from seronegative patients?

Reviewer 2 Report

Authors written a good paper, the experimental design, data analysis, and conclusions are appropriate for the study for the most part. However, the manuscript's impact can be further improved by expanding the points below.

Line 62. 1) Introduction: the contents and the drafting of the general part must be reformed to review the syntax of the topic. Autoimmune disease occurs when the immune system becomes overly active. Authors are invited to integrate this concept by discussing the role of infections in the autoimmune diseases with particular focus on NMOSD and MS disease. Here some interesting papers which might help:

PMID: 31715457; PMID: 30196833; PMID: 29519720

Line 170. Results: better describe the section of results through sub-chapters and identifying well the parts referring to the tables 1, 2 and 3.

Please provide P-values in table 1 and describe in the Figure 1 legend how many patients belonged to each group and which statistical test was used. Moreover, it is not clear what the y-axis represents in this Figure.

Line 221. “Association between cognition, demographic, clinical and patient-centred measures”. Please make a principal component analysis (PCA) and insert the correspond graph in this section.

Line 242. Discussion: authors are invited to integrate the previous concept by discussing the role of infections in the autoimmune diseases with clinical data, inflammation and other major symptoms described in patients with NMOSD.

4 - Updating the references with the papers suggested to in point 1 the article will be more strength in the contents, more attractive to readers and complete.

Best wishes

Author Response

Specific comments

Point 1: Line 62. 1) Introduction: the contents and the drafting of the general part must be reformed to review the syntax of the topic. Autoimmune disease occurs when the immune system becomes overly active. Authors are invited to integrate this concept by discussing the role of infections in the autoimmune diseases, with particular focus on NMOSD and MS disease. Here some interesting papers which might help: PMID: 31715457; PMID: 30196833; PMID: 29519720

Response 1: We thank the reviewer for his/her comments, and we added the following sentences in the discussion (page 8, lines 313-322): “It is also unknown what causes the humoral immune response that produces the AQP4-IgG antibodies. Some infectious agents, even silent infections (Mycobacterium avium subspecies), have been involved in NMOSD aetiology[31,32]. Molecular mimicry between microbes and host peptides have been proposed as a mechanism that would exacerbate autoimmunity and generate autoantibodies. Interestingly, one recent study has shown a different pattern of humoral-driven immune responses against viral agents (retroviruses HERV-W family) between patients with NMOSD compared to patients with MS or MOG-IgG[33]. If such infectious agents and its implication in autoimmunity, could influence cognitive performance, deserve further studies”.

Point 2: Line 170. Results: better describe the section of results through subchapters and identifying well the parts referring to the tables 1, 2 and 3.

Response 2: We agree with the reviewer, and we have added one subchapter on page 4, line 193 (3.1. Demographic, clinical and patient-centred measures of the cohort) and we have renamed the other existing subchapters (3.2. Cognitive characteristics in NMOSD patients; 3.3. Association between cognition, demographic, clinical and patient-centred measures). We have changed “Figure 2” to “Figure 1” on page 6, line 236.

Point 3: Please provide P-values in table 1 and describe in the Figure 1 legend how many patients belonged to each group and which statistical test was used. Moreover, it is not clear what the y-axis represents in this Figure.

Response 3: In Table 1 we described the sample characteristics, and we did not use any statistical comparisons. Instead, in the remaining tables (Table 2, Table 3, Table S1 and Table S2) we showed the p-value after being corrected for multiple comparisons using the false discovery rate (FDR). In the statistical analysis (page 4, lines 189-190), we stated: “We used the false discovery rate (FDR) to correct for multiple comparisons, and we set the significance level at p<0.05”.

We corrected the number in the Figure 1 legend and added the information required for the reviewer in the legend (page 6, lines 236-246): “Figure 1. Cognitive performance in NMOSD patients. (A) Box plots represent the cognitive z-score distribution for each test from the BRB-N in the entire cohort, the x-axis depicts the name of each cognitive test, and the y-axis the z-score for each test. The dotted black horizontal line represents -1.5 SD of the norm. (B) The histograms show the proportions of patients with CP and CI in each cognitive domain. The x-axis shows the names of cognitive domains and the y-axis the number of patients for each domain. The total number of patients in each cognitive domain was 41. Both figures were fitted using R version 3.5.2 (R Foundation for Statistical Computing). SRTS: Selective Reminding Test Long-Term Storage; SRTR: Selective Reminding Test Consistent Long-Term Retrieval; SRTD: Selective Reminding Test Total Delay; SPART: Spatial Recall Test; SPARTD: Spatial Recall Test Delay; SDMT: Symbol Digit Modalities Test; PASAT: Paced Auditory Serial Addition Task; WLG: Word List Generation”.

Point 4: Line 221. “Association between cognition, demographic, clinical and patient-centred measures”. Please make a principal component analysis (PCA) and insert the correspond graph in this section.

Response 4: The principal component analysis (PCA) is a technique for reducing dimensionality by identifying the similarity of the data. As PCA applies a linear transformation that fits the data to a new coordinate system, the use of this method in our study is difficult. We aimed to study the association between cognitive performance with demographic, clinical and mood in patients with NMOSD, and for this, we used linear regression with the Akaike Information Criterion (AIC) to select those variables that best fit the model.

Point 5: Line 242. Discussion: authors are invited to integrate the previous concept by discussing the role of infections in the autoimmune diseases with clinical data, inflammation and other major symptoms described in patients with NMOSD.

Updating the references with the papers suggested to in point 1 the article will be more strength in the contents, more attractive to readers and complete.

Response 5: Following the advice of the reviewer, we added the following sentences in the discussion (page 8, lines 313-322): “It is also unknown what causes the humoral immune response that produces the AQP4-IgG antibodies. Some infectious agents, even silent infections (Mycobacterium avium subspecies), have been involved in NMOSD aetiology[31,32]. Molecular mimicry between microbes and host peptides have been proposed as a mechanism that would exacerbate autoimmunity and generate autoantibodies. Interestingly, one recent study has shown a different pattern of humoral-driven immune responses against viral agents (retroviruses HERV-W family) between patients with NMOSD compared to patients with MS or MOG-IgG[33]. If such infectious agents and its implication in autoimmunity, could influence cognitive performance, deserve further studies”.

Round 2

Reviewer 1 Report

thank you for your response